# Syncytia in Fungi

**DOI:** 10.3390/cells9102255

**Published:** 2020-10-08

**Authors:** Alexander P. Mela, Adriana M. Rico-Ramírez, N. Louise Glass

**Affiliations:** 1Department of Plant and Microbial Biology, University of California, Berkeley, CA 94720, USA; apm8190@berkeley.edu (A.P.M.); amricoramirez@berkeley.edu (A.M.R.-R.); 2Environmental Genomics and Systems Biology Division, Lawrence Berkeley National Laboratory, Berkeley, CA 94720, USA

**Keywords:** syncytia, filamentous fungi, heterokaryon, nucleus, morphology

## Abstract

Filamentous fungi typically grow as interconnected multinucleate syncytia that can be microscopic to many hectares in size. Mechanistic details and rules that govern the formation and function of these multinucleate syncytia are largely unexplored, including details on syncytial morphology and the regulatory controls of cellular and molecular processes. Recent discoveries have revealed various adaptations that enable fungal syncytia to accomplish coordinated behaviors, including cell growth, nuclear division, secretion, communication, and adaptation of the hyphal network for mixing nuclear and cytoplasmic organelles. In this review, we highlight recent studies using advanced technologies to define rules that govern organizing principles of hyphal and colony differentiation, including various aspects of nuclear and mitochondrial cooperation versus competition. We place these findings into context with previous foundational literature and present still unanswered questions on mechanistic aspects, function, and morphological diversity of fungal syncytia across the fungal kingdom.

## 1. Syncytia

Syncytia can be defined as multinucleated cells within a common cytoplasmic environment whose main purpose is to function as a single coordinated unit. There are many cases of syncytia in nature. In animals, for example, the placenta is a multinucleate organ that has an important role in protecting the fetus and transporting nutrients [1]. In muscle cells, multinucleated fibers are formed via the fusion of myoblasts [2], and osteoclasts are multinuclear bone-resorbing cells that are formed via cell fusion [3] (Figure 1a). In the slime mold, *Physarum polycephalum*, individual amoebae fuse to form a large mass of protoplasm containing multiple nuclei [4] that undergo synchronous mitoses [5] (Figure 1b). In plants, syncytia also occur, for example, the endosperm–placental syncytia [6] (Figure 1c). In fungi, diversity of morphological characteristics that have evolved over time, and the syncytial lifestyle may have arisen in hyphal and non-hyphal organisms. For example, chytrids species belong to an ancient phylum of fungi (Chytridiomycota) that diverged at least 750 million years ago from other fungal lineages [7]. Chytrids do not form long multinucleate hyphae that search for nutrients. Instead, they have anucleate ‘rhizoids’ that are derived from a unicellular, multinucleate structure, or more complex rhizomycelia, which share morphological features with filamentous fungal syncytia and multinucleate hyphae.

The hallmark growth habit of filamentous fungi is as multinucleate syncytia (Figure 2a,b). The interconnectedness of fungal syncytia occurs via fusion of germinated asexual spores (germlings) via so-called conidial anastomosis tubes (CATs) [9] or between hyphae within a single colony or between colonies (anastomosis) [10] (Figure 2a–d). Fusion results in the formation of an interconnected network (mycelium) [11,12], which is the foundation for colony establishment and growth. In *Neurospora crassa*, anastomosis is associated with cell-to-cell communication processes that regulate chemotropic growth of germlings and hyphae before cell contact [10]. Germlings/hyphae send and receive signals that guide chemotropic interactions, culminating in cell–cell adhesion, cell wall dissolution, membrane merger, and cytoplasmic mixing [13,14,15,16]. Hyphal anastomosis primarily occurs in the interior of a colony either through hyphal branches or by contact between adjacent hyphae, resulting in the formation of a fusion bridge [10,11]. Hyphal fusion can also occur between colonies with the same or non-identical genotypes [17,18].

The formation of an interconnected syncytium can be beneficial for colony development, facilitating the exchange of genetic material, nutrient transport, improving colony establishment, and increasing colony size [17,19,20,21]. Upon hyphal fusion, cytoplasmic flow can show dramatic changes in directionality [10,22]. Germlings or young colonies can share genetic resources or nutrients through cell fusion, but the process can be restricted as colonies age and undergo hyphal differentiation [17]. The advantages of interconnected syncytial hyphae lie in their ability to transport nutrients through a continuous cytoplasm, allowing the efficient use of the nutrients once a local source has been exhausted [23].

Within a syncytium, heterokaryons can arise due to a spontaneous mutation within a single colony (Figure 3a). In a laboratory strain of *N. crassa*, around 2–3% of the nuclear population harbored mutations that modified the mycelium phenotype, showing the difficulties of maintaining a mycelium with a uniform genetic background [24]. Heterokaryons can also form via germling or hyphal fusion between genetically different cells/colonies, resulting in the coexistence of genetically different nuclei in a common cytoplasm (Figure 2e,f). Heterokaryon formation is an essential element of many fungal life cycles and may be useful to complement mutations. There are also beneficial aspects of the parasexual cycle, such as functional diploidy and mitotic recombination [25] (Figure 3d). However, heterokaryon formation can transmit genetic infections that negatively impact fitness, such as the transmission of mycoviruses [26,27], deleterious mitochondrial DNA, and senescence plasmids [28,29], transposons [30], or parasitic nuclei [31]. In this case, heterokaryon formation and viability are regulated by ‘self/non-self’ or ‘allorecognition mechanisms’ via genetic differences at *het* (heterokaryon) or *vic* (vegetative incompatibility) loci [26,32,33,34,35]. In *N. crassa*, allorecognition and restriction of heterokaryon formation has been dissected and can be summarized in three key points: (i) cells that differ in allorecognition specificity at the *doc* (determinate of communication) loci restrict chemotropic interactions to cells with identical specificity at the *doc* loci [36], (ii) cells that differ in allorecognition specificity at the *cwr* (cell wall remodeling) loci, undergo chemotropic interactions, but cell wall dissolution of adhered cells is blocked [37], and (iii) cells that perform chemotropic interactions and cell fusion undergo a programmed cell death reaction following fusion that restricts cytoplasmic mixing of the two cells/hyphae if they differ in allorecognition specificity at *het* loci [32,38,39,40,41] (Figure 2e–h). Within an *N. crassa* population, a very large number of incompatible genotypes are possible upon segregation of *doc*, *cwr*, and *het* loci, making heterokaryon formation between cells of different genetic backgrounds highly unlikely [41,42].

## 2. Differentiation within Fungal Syncytia

### 2.1. Hyphal Architecture

A fungal syncytium is in contact with the surrounding environment and encounters a variety of external stimuli that require precise and, oftentimes, localized responses. Physiological specialization of hyphae within a colony can partition tasks within the syncytium. Observations of heterogeneous hyphal architecture, such as a morphological transition to thicker diameter ‘trunk hyphae’, differences in hyphal compartment sizes, extension rates, and branching associated with the anatomy of fungal syncytia, have been described [43,44,45,46,47] (Figure 2c). Heterogeneity of syncytial architecture is not limited by hyphal diameter, but also by differences in septation [48] and mobility of organelles, such as nuclei and mitochondria [10,22]. Studies in *Aspergillus niger* showed heterogeneous unidirectional transport of sugar analogs from the colony center to the periphery of the colony, as well as the autonomy of apical compartments (vs. subapical and basal compartments), based on the ability to rapidly plug lysed cells via Woronin bodies [49,50,51]. Recent experiments in the basidiomycete species, *Coprinopsis cinerea*, revealed the presence of specialized trunk hyphae that transported defense response compounds and nutrients bidirectionally across the fungal colony in response to localized fungivory by nematodes, while smaller hyphae exhibited a unidirectional flow [52]. Hyphae can also serve as building blocks for more complex or differentiated hyphal arrangements in basidiomycetes and some ascomycete species, called ‘rhizomorphs’ or ‘mycelial cords’. Rhizomorph is a term used for growing hyphal tips with apical dominance, and mycelial cords are aggregations of up to thousands of younger (and oftentimes melanized) parallel bundled hyphae that are fused by anastomosis to an older, leading hypha [53,54,55]. These rhizomorphs and mycelial cords are highly differentiated and share many structural components that resemble plant roots, such as a small region behind the growth front that undergoes rapid cell division and layers of cells that guard the growing cords against damage by soil particles [56]. These differentiated structures can grow many times faster than normal hyphae and can form linkages between nutrient-poor and nutrient-rich substrates, as well as transporting water and nutrients across large distances.

### 2.2. Colony Aging

The Kingdom Fungi contains some of the longest-living organisms on the planet. The ‘humongous fungus’, or *Armillaria gallica*, is an example of a fungal syncytium that has lived for over 2500 years, covers 37 hectares, and weighs more than 4 × 10^5^ kg [57]. Even through time and distance, the genetic makeup of this single *A. gallica* colony has remained surprisingly stable. The diversity of habitats and time scales in which fungi have colonized the planet suggests that fungal syncytia undergo a variety of morphological changes as the mycelium ages. It has been hypothesized that some iteration of ‘paramorphogenetic compounds’ in the center of the colony accumulate when the mycelium reaches a certain age, and this threshold could be a primary driver of hyphal differentiation spatially and temporally (as opposed to being primarily nutrient-based) [47]. Changes in colony architecture are also associated with young colonies. In *N. crassa*, 22 h following asexual spore germination, branch angles change from 90 degrees to 66 degrees, and by 40 and 44 h, hyphal diameters and extension rates plateau, respectively [47]. In *Ashbya gossypii*, chromosome number and ploidy varies as the colony ages [58]. Cytoplasmic streaming can be prevented in hyphal networks by plugging of septal pores by Woronin bodies or other septal pore proteins, thus preventing cytoplasmic streaming throughout the hyphal network [59,60]. Heterogeneity in the older parts of the colony has been postulated to prevent the systemic spread of pathogens, therefore, maintaining the health of the colony overall [50]. Another process associated with aging colonies is autophagy. This process is integral to recycle nutrients and organelles from older parts of a syncytium to facilitate new growth [61].

One aspect of colony aging is the response to stress in the environment. Although relatively little is known about how syncytium formation directly drives stress response on a cellular or organismal level, there have been several studies suggesting there are benefits of syncytia in dealing with stress. Increased tolerance to variation in ploidy level has been observed in *Ashbya gossypii* cells [62]; homogenization of various nucleotypes and macromolecules also occurs through cytoplasmic flow [22,63]. In addition, compartmentalization of damage and regeneration of growing tips has also been shown to be an important aspect of syncytial morphology in *Aspergillus niger* [51].

### 2.3. Heterogeneity between Hyphal Compartments

The advent of sequencing capabilities and bioinformatics pipelines has made techniques for analyzing subtle differences in transcriptional or translational programs between hyphae possible [64]. The coupling of techniques, such as laser capture microdissection and RNA-seq [65], has been used to isolate specific fungal tissues to assess differences in gene expression. In *Aspergillus niger*, laser capture microdissection and single-cell profiling of hyphal tips in close proximity revealed expression differences between hyphae [66]. Similar experiments between neighboring hyphae revealed heterogeneous secretion of enzymes for carbohydrate acquisition, such as glucoamylase [67]. In *N. crassa*, expression profiling of a sectioned colony from peripheral to internal portions showed spatially distinct mRNA expression patterns. At the periphery of the colony, genes related to polarized growth, biosynthesis of the cell membrane, and cellular signaling were more highly expressed, while for the middle section, genes implicated in energy production showed an increase in expression level [68]; these results were comparable to those found for *A. niger* [69]. These results highlight the fact that in fungal syncytia, different hyphal types occur, and differential gene regulation and important cellular functions can be spatially and temporally regulated across a colony. However, the molecular mechanism and rules whereby this differentiation and differential gene regulation occurs within a syncytium remain obscure.

### 2.4. Nuclear Competition and Cooperation

Genetically distinct nuclei existing in the same cellular space are found in many filamentous fungal syncytia [70,71], although mechanisms governing nuclear coordination and competition in heterokaryons are largely unknown. A long-standing hypothesis is that by sharing a common cytoplasm in syncytia, nuclei would act in concert for the production/utilization of ‘common goods’. Furthermore, heterokaryons containing genetically distinct haploid nuclei would be functionally equivalent to diploid organisms (Figure 3b); expression profiling in the mushroom species, *Agaricus bisporus*, showed nuclear-specific expression patterns that were associated with the nuclei harboring different mating types [72]. In filamentous ascomycete species, genetic diversity within a colony can be generated via parasexual genetics or the transmission of genetic material between nuclei in the absence of mitosis or meiosis (Figure 3d). The parasexual cycle has been observed in many haploid filamentous fungi and is routinely used for genetic linkage mapping [25,73,74]. Despite incompatibility reactions between strains that affect the ability to form viable heterokaryons, under certain circumstances, these internuclear genetic interactions provide additional chances for mutually beneficial DNA elements to be transferred and utilized by genetically distinct nuclei for the benefit of the syncytium. ‘Accessory’, or ‘dispensable chromosomes’, often contain virulence genes, and through chromosomal rearrangements, spontaneous pathogenic strains can arise from otherwise non-pathogenic isolates [75]. In studies of *Fusarium* spp. and *Nectria haematococca*, heterokaryotic syncytia were shown to exchange and retain chromosomes/genes between nuclei associated with virulence, while in some cases eliminating those not involved in pathogenicity [76,77,78,79]. In *Saccharomyces cerevisiae* cells, which do not usually form heterokaryons, a small portion of the mating population (~1%) may form a transient type of heterokaryon, termed a ‘cytoductant’. Cytoductants undergo conventional DNA and mitochondrial genetic recombination as seen in normal zygotes, but also show chromosome transfer, which suggests that this could be a fundamental property of fungi that plays a role in providing genetic variation in a population [80].

One caveat with these concepts is that nuclei within syncytia also potentially undergo competition, whereby a nucleus with a clear fitness advantage could dominate over less advantaged nuclei throughout a heterokaryotic colony [12] (Figure 3c). A major technical issue for filamentous fungal researchers tackling these questions is the identification and visualization of genetically distinct nuclei within syncytia. Fluorescent dyes and fluorophore-tagged nuclear proteins are often readily exchanged between nuclei [22,81,82]. Previous studies with filamentous ascomycete species showed that environmental pressures can drive unbalanced nuclear ratios [83,84]. Systematic studies using microsatellite markers for the analysis of unbalanced nuclear ratios in basidiomycete species also revealed that the representation of a genotype can be augmented by environmental pressures [85,86]. In many heterokaryon combinations, one nucleus dominated in terms of representation, suggesting inherent genetic fitness, in addition to environmental pressure, can drive heterogeneity of nuclear ratios within a syncytium. Mutations in a subset of nuclei within a homokaryon can also create heterokaryons, and those mutations can be beneficial or unfavorable to the fitness of the syncytium [85,87]. In some cases, syncytia may contain ‘senescent nuclei’, which despite imparting clear morphological defects and being detrimental to the colony as a whole, they are able to over proliferate, either due to faster replication or by other means [88,89]. These data suggest additional levels of regulation could contribute to how genotypic autonomy is regulated, such as the time a nucleus (or mitochondria) spends in a transcriptional versus replicative state.

Access by nuclei to asexual or sexual spores during conidiation or before meiosis could also promote nuclear competition [90]. Recent work with natural mating-type heterokaryotic species, *Neurospora tetrasperma,* showed a clear bias for selection by mating-type (*mat A*) and associated genes during vegetative growth and asexual development, although ratios of *mat A* and *mat a* nuclei equalized during sexual development [91]. Heterokaryons of the basidiomycete species *Heterobasidion parviporum* consisting of sibling-composed nuclei tended to produce nuclear distributions and germination rates of asexual spores that were very similar to homokaryons. However, heterokaryons composed of non-sibling related nuclei tended to form uninucleate conidia that germinated faster than those that were bi- and trinucleate from non-sibling heterokaryons.

### 2.5. Nuclear Autonomy

Nuclei maintain autonomy during cellular events that take place in a common cytoplasm in filamentous fungi, oftentimes at relatively short internuclear distances, such as asynchronous mitosis [92,93] and parasynchronous mitosis [94,95] (Figure 3e). A number of hypotheses for asynchronous cell cycle progression have been proposed in *A. gossypii*: (i) subsets of nuclei could emit cell cycle blocking molecules, such as CDK inhibitors, locally restricting cell cycle progression of neighboring nuclei; (ii) seemingly random distribution of factors which positively promote cell cycle progression, in addition to varying requirements for each nucleus to enter the cell cycle may lead to asynchronicity; (iii) positioning of nuclei in relation to cortical markers, which may exhibit very specific localization, could determine the fate of nuclei to enter the cell cycle; (iv) nuclei and associated transcripts/proteins may be separated in the cytoplasm, for example, organelles or membranes could partition translated proteins to a specific part of a nucleus or subset of nuclei; (v) cytoskeletal elements, nuclear pore proteins, or transcription factors may be asymmetrically distributed between ‘mother’ and ‘daughter’ nuclei upon mitotic exit, thereby mis-synchronizing future cell cycle timing [93].

Understanding is also still lacking about the positioning and fate of mRNA transcripts and proteins within a syncytial cell, relative to the nucleus of origin. Recent studies involving microtubule-associated nuclear repulsion [96], as well as cytoplasmic streaming and ‘eddy currents’ [59] suggest that ‘nuclear neighborhoods’ occur in multinucleate hyphae, and microenvironments in the cytoplasm regulate where nuclei, organelles, and cytoskeletal elements aggregate, affect transcriptional patterns and access local pools of ‘common goods’. These ‘neighborhoods’ could be partially due to a shift in the physical state of the cytoplasm to a gel-like conformation (Figure 3g), mediated by a phase separation via unstructured regions of polypeptides, as has been shown in vitro [97,98,99]. In *A. gossypii*, mechanistic details connecting the cell cycle and nuclear coordination were assisted by adapting single-molecule fluorescence in-situ hybridization (smFISH) protocols to visualize mRNA transcripts of mitotic cyclins in fixed cells. PolyQ-driven assemblies of protein-RNA were found to affect the spatial distribution of transcripts for the cell cycle regulator *CLN3* by facilitating a shift in physical state, followed by aggregation with complexes of similar physical properties, both in-vivo and in-vitro [100,101,102,103]. These observations point to how the partitioning of the nucleoplasm and cytoplasm can facilitate asynchronous and parasynchronous cellular events (Figure 3e).

The ratio of nuclei to cytoplasm also seems to vary across the fungal kingdom. In *A. gossypii*, it has been shown that the ratio of nuclei per unit volume of cytoplasm (#N/C) remains constant even when nuclei are clustered, as is the case with the mutant defective in the function of dynactin (*jnm1Δ*) [102]. Dynein mutants in several fungal species lead to clusters of nuclei, indicating a role for dynein in nuclear distribution and migration in hyphae [104,105,106]. Nuclear repulsion facilitates sufficient spacing between nuclei to allow for a stable cytoplasmic–nuclear ratio throughout growth (Figure 3f), despite exhibiting asynchronous mitoses [93,96]. Upon removal of nocodazole, which blocks nuclear division, nuclei underwent rapid divisions to reestablish wildtype-level internuclear spacing and cytoplasmic–nuclear ratios [96]. In contrast, although *N. crassa* also exhibit asynchronous mitoses, fixed internuclear distances are not maintained throughout growth [22]. Slime molds like *P. polycephalum*, which share many similar properties with fungal syncytia, undergo synchronous mitosis, where the entire nuclear and cytoplasmic contents double together [107]. *Aspergillus nidulans* undergoes a parasynchronous mitotic wave, where the more apical compartments undergo rapid extension accompanied by faster nuclear division, while more basal compartments remain mitotically inactive or less active [108]. In this case, branching was correlated with nuclear division and an increase in cytoplasmic volume. In *Schizosaccharomyces pombe*, the cell maintains a nuclear size proportional to cell size (N/C ratio) [109]. Mutants with defects in nucleocytoplasmic mRNA transport and lipid synthesis were altered in their N/C ratio, indicating that cells must regulate nucleocytoplasmic transport and nuclear membrane growth to maintain appropriate N/C ratios within the cell [110]. Much remains to understand the regulatory mechanisms associated with nuclear autonomy and its integration with cellular processes, in particular, how unicellular and syncytial fungi ‘sense’ and respond to fluctuations in cytoplasmic volume and nuclear content under different environments and growth habits.

### 2.6. Mitochondrial Autonomy

Although the nuclear genome is thought of as being the primary source of genomic information in the cell, mitochondria also contain a distinct genome. In filamentous fungi, the transmission of mitochondria solely by one parent (in ascomycetes, this is primarily the ‘maternal’ parent, regardless of mating-type) can occur during sexual reproduction [112,113,114]. In *N. tetrasperma*, mitochondria show a ‘maternal’-only inheritance via fertilization with specialized mating hyphae (trichogynes). However, in cases where mating was initiated by hyphal fusion, one mitochondrial DNA fully replaced the mitochondrial DNA of the hyphal fusion partner [113]. In heterothallic basidiomycete species, there is an ordered exchange of nuclei, but not cytoplasm between vegetative cells of monokaryons, creating a dikaryon with a homogenous distribution of two genetically distinct nuclei per cellular compartment, whereas a homogenous or heterogeneous mix of mitochondrial genomes can occur. This creates a ‘patchwork’ of mitochondrial genotypes throughout the mycelial dikaryon [115].

## 3. Conclusion and Outlook

Despite decades of research on fungal syncytia, we have only just begun to elucidate many mechanistic details of how they function and proliferate. There appears to be far more nuclear and mitochondrial autonomy and natural partitioning of cellular compartments than previously hypothesized within fungal syncytia. Despite the interconnectedness of filamentous fungal networks, researchers have unveiled a picture of a more heterogeneous landscape of gene expression, metabolic function, and protein production, both spatially and temporally, within growing syncytia. Future work in this area of research should address some of the unanswered questions, such as (i) What are the basic rules that govern fungal syncytia formation and viability? (ii) How and to what extent do genetically distinct nuclei and mitochondria coordinate and/or compete with one another within a fungal syncytium? (iii) what internal/external stimuli govern differentiation into particular cell types as a colony ages? (iv) Evolutionarily-speaking, how and why did the transition from unicellularity to a syncytial lifestyle develop across the Kingdom Fungi? Considering filamentous fungi are currently used as the ‘workhorses’ of modern industrial production of compounds, and there has been a multitude of fungal-derived materials and bio-products manufactured in recent years, understanding these questions has implications for engineering syncytial fungi for optimal protein production in industrial settings. Future studies in this field will also improve our understanding of how large hyphal networks (>15 hectares) affect carbon cycling and nutrient translocation in the environment.

## Figures and Tables

**Figure 1 cells-09-02255-f001:**
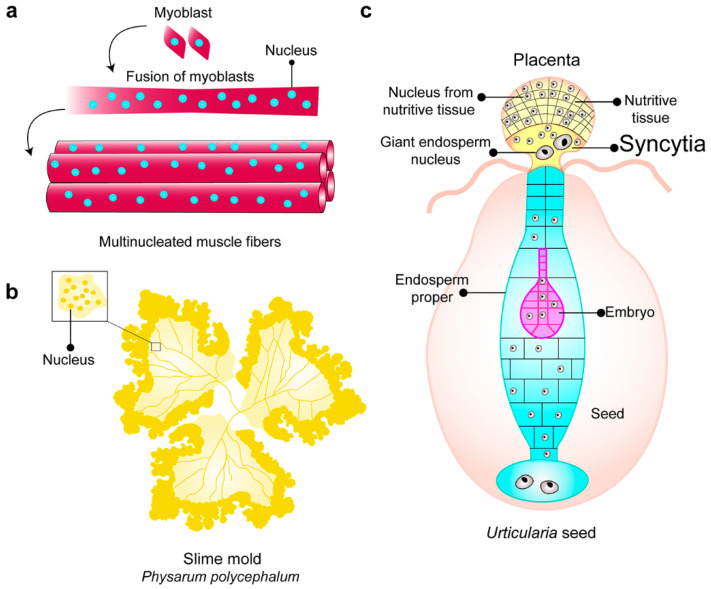
Examples of syncytia in nature. (**a**) Myoblasts are fused to form multinucleated muscle fibers. (**b**) Slime mold *Physarum polycephalum*, multinucleate protoplasm formed by the fusion of individual amoebae. (**c**) Endospermal–placental syncytia in developmental stages of *Urticularia* seeds, the formation of syncytia occurs in the placenta. The syncytia harbor two populations of nuclei, a nucleus from the nutritive tissue, and a giant endosperm nucleus (modified from [8]).

**Figure 2 cells-09-02255-f002:**
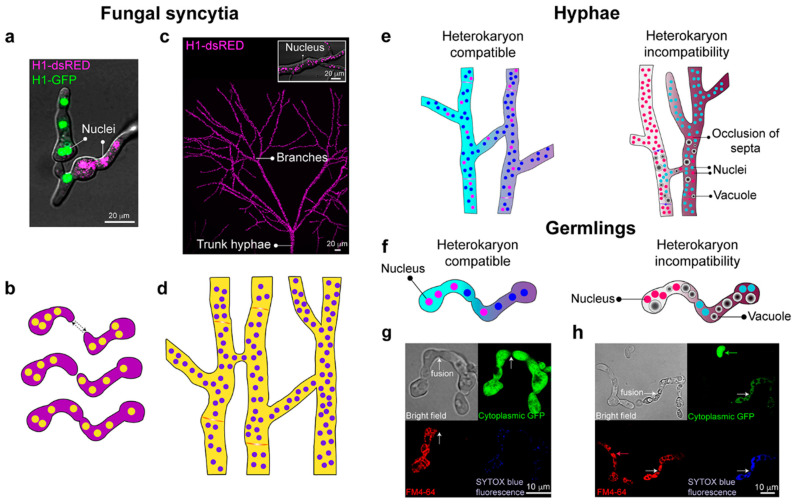
Formation of interconnected fungal syncytia. (**a**) The fusion of compatible strains of *Neurospora crassa*, whose nuclei are labeled with either histone H1-GFP or H1-dsRED (green and magenta, respectively). (**b**) Genetically identical germs grow to each other (dashed arrows), fuse, and give rise to interconnected multinucleate syncytia. (**c**) Heterogeneity of the mycelial network and syncytia formation. Magenta dots are nuclei marked with histone H1-DsRed. The box shows a close-up of a hypha, showing the marked nuclei. (**d**) Hyphal fusion within a colony contributes to an interconnected syncytium. (**e**) In hyphae, heterokaryon formation can occur when there are no differences at *het* (heterokaryon) or *vic* (vegetative incompatibility) loci. In contrast, genetic differences at these loci result in heterokaryon incompatibility, which triggers compartmentalization of the fusion compartment due to occlusion of the septum, vacuolization of the hyphae, and eventual cell death. (**f**) In germlings, heterokaryon formation can occur when there are no differences at the *rcd-1* (regulator of cell death-1) and *plp-1/sec-9* (patatin-like phospholipase-1) loci. In contrast, differences at these loci result in heterokaryon incompatibility, rapidly triggering a cell death reaction that is a similar process in hyphae [38,39]. (**g**) Micrographs show the fusion of compatible germlings. One of the germlings is marked with cytoplasmic Green Fluorescent Protein (GFP) (green) and has undergone cell fusion with a compatible germling stained with FM4-64 fluorescent dye (red). Fusion is evident by the fact that GFP fluorescence can be observed in both germlings due to cytoplasmic mixing, and cell death does not occur, as indicated with the absence of SYTOX Blue fluorescence (death cell stain). (**h**) Cell fusion between germlings with genetic differences at the *plp/sec-9* loci results in rapid cellular vacuolization and death, as demonstrated with the staining of SYTOX Blue fluorescence. White arrows indicate fusion events. Micrographs also show two germlings that have not undergone cell fusion and are healthy (green; GFP and red arrows; FM4-64). Micrographs (**g**,**h**) courtesy of Dr. Jens Heller (UCB Glass Laboratory).

**Figure 3 cells-09-02255-f003:**
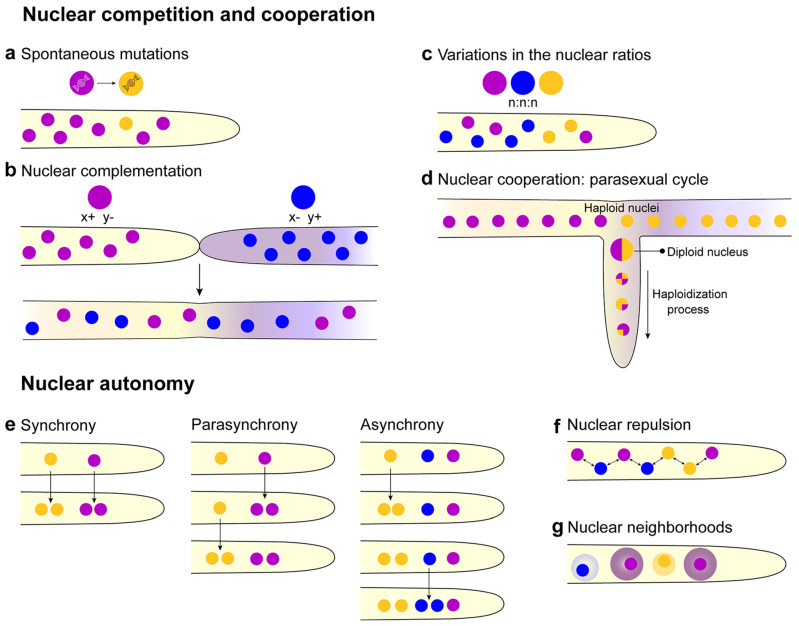
Nuclear patterns in multinucleate syncytia. (**a**) Spontaneous mutation in nuclei can give that nucleus an advantage or disadvantage over the rest of the nuclei in the syncytial population. (**b**) Nuclear complementation can occur if a nucleus lacks a gene (x− or y−) encoding a function necessary for survival. x− or y− function can be complemented by the presence of a second nucleus, which is functional for that gene (x+ or y+). Complementation between nuclei in a syncytium can occur with spontaneous mutations (**a**) or via complementation upon fusion with another individual that can produce the missing component. (**c**) In multinucleate syncytia, variations in the nuclear ratios can occur, where one nucleus can dominate. (**d**) Generation of nuclear heterogeneity through the parasexual cycle. Haploid nuclei in a heterokaryon, formed either by spontaneous mutation or via fusion with a different strain, undergo karyogamy to form a heterozygous diploid nucleus. Repeated rounds of mitotic recombination and mitotic nondisjunction result in loss of chromosomes, producing haploid nuclei with unique genotypes. (**e**) Different patterns of nuclei division in syncytia. Synchrony: all the nuclei divide at the same time. Parasynchrony, the mitosis is initiated in one nucleus, and then linearly, the adjacent nucleus starts to divide after the first one. Asynchrony, the nuclei divide independently of each other (modified from [93]. (**f**) The nuclei show repulsion to delimit their cytoplasmic territory [96], and within a hypha, a regular number of nuclei per unit volume of cytoplasm (#N/C) is observed [102]. (**g**) Nuclear neighborhoods can be organized around nuclei that affect cell cycle regulation and, potentially, other regulatory processes (modified from [111]).

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
