# Peer review of "Syncytia in Fungi"

_cells, 2020, doi:10.3390/cells9102255_

Round 1

Reviewer 1 Report

This is truly a wonderful review on the multinucleate fungal syncytia. It covers important aspects of growth and differentiation of fungal colonies, especially the behaviors of nuclei in multinucleate syncytia. It will be very helpful to any reader who is interested in the current knowledge of fungal syncytia. I only have a few minor points for the authors to consider before publication.

  1. Figure 3f legend, "Hyphae maintain a constant nuclear to cytoplasmic (N/C) ratio via nuclear repulsion." Is there a good reference for this statement?  It may be better to point out that this happens in normal hyphae. I remember Amy Gladfelter's lab found that even when nuclei are clustered together in a jnm1 mutant defective in dynein function (Jnm1 encodes a protein in the dynactin complex), the N/C ratio is still maintained (Dundon et al., 2016 MBoC).
  2. It would be better to mention that dynein is required for nuclear migration/distribution in fungal hyphae. Although cytoplasmic streaming plays a role in nuclear movement, dynein mutants in several fungal species all exhibit a severe nuclear clustering phenotype (Plamann et al 1994, Xiang et al., 1994; Alberti-Segui et al., 2001).
  3. Line 207, "A major technical issue for filamentous fungal researchers tackling these questions is the identification and visualization of genetically distinct nuclei within syncytia. Fluorescent dyes and fluorophore-tagged nuclear proteins are often readily exchanged between nuclei (20, 76, 77)." Great point! It would be better if you could add somewhere (may be as a future direction) experiments that can potentially be tried in filamentous fungi to solve this problem. Would it work if one labels the dividing nuclei in one strain with EdU before mixing the strain with another strain and then analyze the fates of these nuclei in relation to the other nuclei in the heterokaryon by using the Click-iT EdU kit as well as general nuclei staining? In this context, I actually also wonder how strong the evidence is for nuclear competition based on previous papers.
  4. Line 270, "A. nidulans" should be changed to "Aspergillus nidulans" because this is the only time you mention this species.

Author Response

Reviewer 1

This is truly a wonderful review on the multinucleate fungal syncytia. It covers important aspects of growth and differentiation of fungal colonies, especially the behaviors of nuclei in multinucleate syncytia. It will be very helpful to any reader who is interested in the current knowledge of fungal syncytia. I only have a few minor points for the authors to consider before publication.

We thank the reviewer for their support of our review.

  1. Figure 3f legend, "Hyphae maintain a constant nuclear to cytoplasmic (N/C) ratio via nuclear repulsion." Is there a good reference for this statement?  It may be better to point out that this happens in normal hyphae. I remember Amy Gladfelter's lab found that even when nuclei are clustered together in a jnm1 mutant defective in dynein function (Jnm1 encodes a protein in the dynactin complex), the N/C ratio is still maintained (Dundon et al., 2016 MBoC).

Thank you for your comment. We realized that the figure 3f legend is confusing, so we rewrote it as follows:

The nuclei show repulsion to delimit their cytoplasmic territory [95] and within a hypha a regular number of nuclei per unit volume of cytoplasm (#N/C) is observed [101].

Additionally, we include lines in the main text:

In A. gossypi, it has been shown that the ratio of nuclei per unit volume of cytoplasm (#N/C) remains constant even when nuclei are clustered, as is the case with the mutant defective in the function of dynactin (jnm1D)” [101]. Dynein mutants in several fungal species leads to clusters of nuclei, indicating a role for dynein in nuclear distribution and migration in hyphae [103-105].

  1. It would be better to mention that dynein is required for nuclear migration/distribution in fungal hyphae. Although cytoplasmic streaming plays a role in nuclear movement, dynein mutants in several fungal species all exhibit a severe nuclear clustering phenotype (Plamann et al 1994, Xiang et al., 1994; Alberti-Segui et al., 2001).

We welcome this suggestion, and it has been added in the main text shown above:

  1. Line 207, "A major technical issue for filamentous fungal researchers tackling these questions is the identification and visualization of genetically distinct nuclei within syncytia. Fluorescent dyes and fluorophore-tagged nuclear proteins are often readily exchanged between nuclei (20, 76, 77)." Great point! It would be better if you could add somewhere (may be as a future direction) experiments that can potentially be tried in filamentous fungi to solve this problem. Would it work if one labels the dividing nuclei in one strain with EdU before mixing the strain with another strain and then analyze the fates of these nuclei in relation to the other nuclei in the heterokaryon by using the Click-iT EdU kit as well as general nuclei staining? In this context, I actually also wonder how strong the evidence is for nuclear competition based on previous papers.

EdU and Click-iT EdU kit are excellent suggestions. We believe that this assay could potentially be useful, but should be tested first. We have not included this suggestion in the text, as it is not within the main objective of this review. 

  1. Line 270, "A. nidulans" should be changed to "Aspergillus nidulans" because this is the only time you mention this species.

Corrected.

Reviewer 2 Report

In this short review, the authors focus on the importance of syncytia to the lifestyle of filamentous fungi. They initially define syncytia and place fungal syncytia in the context of those found in other systems. They describe the mechanisms by which syncytia are generated, including hyphal anastomosis and germling fusion. Characteristic features of fungal syncytia are described, including differentiation within mycelia as well as hyphae compartmentation. The relationship between nuclei within a syncytium is also discussed, along with a short section on mitochondrial autonomy. Overall, this is a highly informative review that summarizes several key concepts underlying the formation, maintenance, and function of fungal syncytia. It should provide useful background for interested fungal researchers, and also provides a broader comparative perspective that will be of value to those who study syncytia in other systems.

The following comments are provided for the authors to consider;

1. Perhaps one of the fundamentally important roles of hyphae syncytia is the proper management of stress responses. For example, compartmentalization of damage and regeneration of growing tips is presumably important, as would be the ability to segregate damaged nuclei to limit their effects on growth and development. Although there is relatively little that is known about how these processes are regulated, the authors should at least note their potential importance and summarize what is known.

2. The authors do comment briefly on the evolution of fungal syncytial (line 306). In this context, they should consider a recent article by Laundon et al. (2020)(PMID: 32517626) that outlines the mechanisms involved in morphogenesis of chytrid rhizoids and suggests broad similarities to hyphae organization. In a sense, syncytial may have existed ion early non-hyphae chytrids.

3. Although I understand the rationale behind use of the term "rules" (e.g., lines 9, 302), the varied approaches that underlie syncytium formation suggest that there might be no hard and fast rules (i.e., they might be very general). It would be useful if the authors attempted to organize in a summary table the rules that can be discerned based on that is already known.

4. Line 204. The authors might note that chromosome transfer has been observed between the two nuclei of yeast heterokaryons (cytoductants), which suggests that this is a fundamental property of fungi that undoubtedly plays a key role in generating variation.

Author Response

Reviewer 2.

In this short review, the authors focus on the importance of syncytia to the lifestyle of filamentous fungi. They initially define syncytia and place fungal syncytia in the context of those found in other systems. They describe the mechanisms by which syncytia are generated, including hyphal anastomosis and germling fusion. Characteristic features of fungal syncytia are described, including differentiation within mycelia as well as hyphae compartmentation. The relationship between nuclei within a syncytium is also discussed, along with a short section on mitochondrial autonomy. Overall, this is a highly informative review that summarizes several key concepts underlying the formation, maintenance, and function of fungal syncytia. It should provide useful background for interested fungal researchers, and also provides a broader comparative perspective that will be of value to those who study syncytia in other systems.

We thank the reviewer for their support of our review.

  1. The following comments are provided for the authors to consider;

Perhaps one of the fundamentally important roles of hyphae syncytia is the proper management of stress responses. For example, compartmentalization of damage and regeneration of growing tips is presumably important, as would be the ability to segregate damaged nuclei to limit their effects on growth and development. Although there is relatively little that is known about how these processes are regulated, the authors should at least note their potential importance and summarize what is known.

We thank the reviewer for this excellent suggestion and have included the following text in the review.

One aspect of colony aging is response to stress in the environment.  Although relatively little is known about how syncytium formation directly drives stress response on a cellular or organismal level, there have been several studies suggesting there are benefits of syncytia in dealing with stress.  Increased tolerance to variation in ploidy level has been observed in Ashbya gossypii cells [61]; homogenization of various nucleotypes and macromolecules also occurs through cytoplasmic flow [21, 62]. In addition, compartmentalization of damage and regeneration of growing tips has also been shown to be an important aspect of syncytial morphology in Aspergillus niger [50].

  1. The authors do comment briefly on the evolution of fungal syncytial (line 306). In this context, they should consider a recent article by Laundon et al. (2020)(PMID: 32517626) that outlines the mechanisms involved in morphogenesis of chytrid rhizoids and suggests broad similarities to hyphae organization. In a sense, syncytial may have existed ion early non-hyphae chytrids.

Thank you for the suggestion. The following has been added.

In fungi, diversity of morphological characteristics that have evolved over time, and the syncytial lifestyle may have arisen in hyphal and non-hyphal organisms.  For example, chytrids species belong to an ancient phylum of fungi (Chytridiomycota) that diverged at least 750 mya from other fungal lineages [7]. Chytrids do not form long multinucleate hyphae that search for nutrients, instead they have anucleate ‘rhizoids’ that are derived from a unicellular, multinucleate structure, or more complex rhizomycelia, which share morphological features with filamentous fungal syncytia and multinucleate hyphae.”                                                                                       ‘

  1. Although I understand the rationale behind use of the term "rules" (e.g., lines 9, 302), the varied approaches that underlie syncytium formation suggest that there might be no hard and fast rules (i.e., they might be very general). It would be useful if the authors attempted to organize in a summary table the rules that can be discerned based on that is already known.

Although we use the term “rules” to describe the approaches that underlie syncytium formation, we concur that there may not be hard and fast rules.  We would like to avoid specifically stating these rules, as they may be more plastic and worry that these may become ‘dogma’ in the literature if plainly outlined as solid rules. 

  1. Line 204. The authors might note that chromosome transfer has been observed between the two nuclei of yeast heterokaryons (cytoductants), which suggests that this is a fundamental property of fungi that undoubtedly plays a key role in generating variation.

We thank the reviewer for this suggestion. The following have been added.

In Saccharomyces cerevisiae cells, which do not usually form heterokaryons, a small portion of the mating population (~1%) may form a transient type of heterokaryon, termed a ‘cytoductant’.  Cytoductants undergo conventional DNA and mitochondrial genetic recombination as seen in normal zygotes, but also show chromosome transfer, which suggests that this could be a fundamental property of fungi that plays a role in providing genetic variation in a population [79].